# How Does Vitamin D Affect Immune Cells Crosstalk in Autoimmune Diseases?

**DOI:** 10.3390/ijms24054689

**Published:** 2023-02-28

**Authors:** Daniela Gallo, Denisa Baci, Natasa Kustrimovic, Nicola Lanzo, Bohdan Patera, Maria Laura Tanda, Eliana Piantanida, Lorenzo Mortara

**Affiliations:** 1Endocrine Unit, Department of Medicine and Surgery, University of Insubria, ASST dei Sette Laghi, 21100 Varese, Italy; 2Immunology and General Pathology Laboratory, Department of Biotechnology and Life Sciences, University of Insubria, 21100 Varese, Italy; 3Molecular Cardiology Laboratory, IRCCS-Policlinico San Donato, San Donato Milanese, 20097 Milan, Italy; 4Center for Translational Research on Autoimmune and Allergic Disease—CAAD, Università del Piemonte Orientale, 28100 Novara, Italy

**Keywords:** autoimmune diseases, immune cells, vitamin D, vitiligo, autoimmune thyroid disorders, multiple sclerosis

## Abstract

Vitamin D is a secosteroid hormone that is highly involved in bone health. Mounting evidence revealed that, in addition to the regulation of mineral metabolism, vitamin D is implicated in cell proliferation and differentiation, vascular and muscular functions, and metabolic health. Since the discovery of vitamin D receptors in T cells, local production of active vitamin D was demonstrated in most immune cells, addressing the interest in the clinical implications of vitamin D status in immune surveillance against infections and autoimmune/inflammatory diseases. T cells, together with B cells, are seen as the main immune cells involved in autoimmune diseases; however, growing interest is currently focused on immune cells of the innate compartment, such as monocytes, macrophages, dendritic cells, and natural killer cells in the initiation phases of autoimmunity. Here we reviewed recent advances in the onset and regulation of Graves’ and Hashimoto’s thyroiditis, vitiligo, and multiple sclerosis in relation to the role of innate immune cells and their crosstalk with vitamin D and acquired immune cells.

## 1. Introduction

Vitamin D (VitD) is a secosteroid hormone that is primarily involved in bone health and mineral metabolism [1]. Since its discovery, mounting evidence has supported broader actions of VitD, including the regulation of cell proliferation and differentiation [2,3]. The involvement of VitD in immune system functions stems from studies showing the presence of the VitD receptor (VDR) in T and B cells, monocytes, macrophages, dendritic cells (DCs), and natural killer (NK) cells. These cells also express the 1α-hydroxylase enzyme (CYP27B1), which is responsible for the activation of VitD precursors (25(OH)VitD) to calcitriol [1,4,5]. Thus, VitD status affects the activation, proliferation, and phenotype of immune cells, which, in turn, can directly activate VitD precursors (Figure 1). These preliminary results inspired epidemiological/observational studies in humans, the majority of which revealed an opposite association between VitD levels and the risk of developing chronic diseases, including autoimmune disorders [6]. Mendelian randomized trials supported the correlation between a higher risk of several autoimmune disorders and a poor VitD status, as determined by polymorphisms of genes involved in VitD activation/transport/local availability [7]. Recently, supplementation with VitD (2000 IU/day) in a placebo-controlled trial (ClinicalTrials.gov NCT01351805 and NCT01169259) resulted in a 22% reduction in the incidence of autoimmune diseases, including autoimmune thyroid disorders (AITDs) and psoriasis, with a median follow-up of 5.3 years [8].

About 5–8% of the general population suffers from autoimmune diseases [9], which develop due to the interaction of genetic/epigenetic, endogenous, and environmental factors [10,11]. Three key stages were described, namely, self-tolerance loss, reduced percentage/activity of regulatory immune cells, and environmental/endogenous precipitating factors [10]. So far, genome-wide association studies (GWAS) identified numerous risk loci for autoimmunity [12,13] that were principally linked to classical human leukocyte antigens (HLA) alleles. Among non-HLA candidate genes, variants of regulatory pathways (e.g., nuclear factor (NF)-kB) and factors related to immune response (e.g., transcription factors forkhead box P3 (FOXP3), BTB domain and CNC homolog 2 (BACH2), and ribonuclease T2 (RNASET2)) are shared by different autoimmune disorders, including MS and AITDs [13,14].

Overall, almost 8 out of 10 patients with autoimmune diseases are women [15,16]. As a response to infections, vaccinations, or traumas, a greater antibody production was described in female subjects, while a stronger inflammatory reaction was observed in men [15]. Since the X chromosome holds genes involved in immune processes, partial X chromosome inactivation or reactivation in females might lead to the overexpression of genes such as FOXP3, chemokine receptor (CXCR3), toll-like receptor (TLR) 7 and 8, CD40 ligand, and interleukin (IL)-9 receptor [17,18,19,20,21]. This hypothesis is supported by the increased susceptibility to autoimmune diseases observed in syndromes linked to aneuploidies, such as a missing X chromosome in women with Turner syndrome or extra X chromosomes in men with Klinefelter syndrome [18]. Central and peripheral immune regulation is significantly affected by circulating sex hormone levels and their intracellular and extracellular receptors [22,23,24,25]. Estrogen receptors were described as present in most immune cells, while androgen receptors were identified in T and B lymphocytes, DCs, neutrophils, and macrophages [26]. Overall, estrogens enhance humoral immunity, while testosterone and progesterone could exert immunosuppressive effects by interacting with receptors on innate and adaptive immune cells, non-genomic pathways, and epigenetic modifications [26,27,28,29]. Estrogens skew the immune response toward a Th2-phenotype by stimulating the release of anti-inflammatory cytokines (interleukins 4 and 10 and tumor growth factor (TGF)-β) and promoting Treg cells activation [17,20,30,31,32], but can also stimulate antibodies production and allow autoreactive lymphocyte survival to interfere with thymic negative selection [17,31,33]. In murine models of autoimmune disorders, females showed higher disease prevalence and severity compared with males [26]. More specifically, in the autoimmune thyroiditis mouse model, estrogens could stimulate antibody production and disease severity, with decreased risk after female castration; in contrast, male castration might increase antibody production and disease incidence [15].

This review focused on the implications of VitD in the regulation of the immune processes that ultimately result in clinically apparent autoimmune diseases. We chose to better detail AITDs, vitiligo, and multiple sclerosis, which might concur in the context of “Autoimmune Polyendocrine Syndromes” (APS) type III class C (APS3C). Instances of APS3C are relatively frequent and are defined by the combination of an AITD and another organ-specific autoimmune disease (except adrenal insufficiency) involving, for example, the integumentary system (e.g., vitiligo, psoriasis, and alopecia) and nervous system (e.g., myasthenia gravis and multiple sclerosis) [34].

## 2. Vitamin D and Immune Regulation

Active VitD (calcitriol or 1,25(OH)2D3) regulates the transcription of more than 300 genes in humans, including some involved in the immune system’s functions, by interacting with the complex vitamin D receptor (VDR)/retinoic acid/promoter. The ability of VitD to boost the innate response and suppress the adaptive ones is supported by preclinical studies. Early clinical evidence comes from the application of cod liver oil supplementation in the prevention/treatment of tuberculosis [35]. Calcitriol promotes the differentiation of monocytes into macrophages and stimulates their antimicrobial activity by increasing the transcription of antimicrobial peptide genes, such as beta-defensin and cathelicidin antimicrobial peptide [35,36]. Small et al. recently discovered that monocytes and monocyte-derived macrophages exposed to calcitriol increased the expression of the pro-phagocytic molecule “complement receptor of the immunoglobulin family” (CRIg) and that this effect is amplified by inducing the transcription of the VDR [37]. Notably, the expression of CYP27B1 in immune cells is regulated by pathogens and pathogen-related inflammatory pathways, independently from calcium homeostasis, thus supporting the anti-microbial role of VitD [38,39,40]. These discoveries paved the way for speculations on the potential protective role of VitD treatment against SARS-CoV-2 infection [41].

In experimental models, calcitriol reduces monocytes’ synthesis of inflammatory cytokines, such as tumor necrosis factor (TNF)-α, IL-1, IL-6, IL-8, and IL-12. Calcitriol also encourages the adoption of a tolerogenic phenotype by dendritic cells (DCs) by increasing inhibitory molecules (e.g., immunoglobulin-like transcript 3 and programmed death ligand) [42,43,44]. The derived “tolerogenic” DCs are characterized by the low expression of molecules (CD80, CD86, and CD40) and cytokines (IL-12 and IL-23) involved in antigen presentation and lymphocyte activation. Thus, the interaction between tolerogenic DCs and T cells induces T cell anergy and apoptosis and promotes the synthesis of tolerogenic cytokines (IL-10) [45]. In peculiar conditions, toll DCs may promote lymphoid polarization to anti-inflammatory subsets, such as antigen-specific T regulatory cells (Treg cells), by inducing FOXP3 transcription [46]. Treg cells dampen the response of effector T cells by releasing inhibitory cytokines (e.g., IL-10 and TGF-β), granzymes, and perforin [47,48], and via the expression of inhibitory co-receptors, such as CTLA-4, which prevent antigen presentation [47]. Thus, Treg cells are pivotal in maintaining immune tolerance to self-antigens. Numerous animals and in vitro studies showed that VitD promotes Treg cell induction [47,48].

In vitro, high VitD levels directly suppress T cell proliferation and cell cycle progression from gap (G) 1a to G1b phases and promote the shift from T helper (Th)-1/Th17 to Th2 phenotype by suppressing Th1 (IL-2, interferon-gamma (IFNγ), TNF-α) and Th17 (IL-17, IL-21) cytokines in favour of Th2 (IL-4, IL-5, IL-9, IL-13) cytokines [49,50].

VitD inhibits B cell activation and differentiation to plasma cells and induces apoptosis of activated B cells and immunoglobulin synthesis [46].

In this context, the relationship between VitD and IL-33 could be of interest because these two molecules share some signaling pathways. For instance, both IL-33 and VitD encourage the activation of Th2 while inhibiting the Th1 response and they both orchestrate immunoregulatory cells [51]. In some biological processes, they act in synergy, while in others, they act by controlling and modulating each other [52]. Nevertheless, the clarification of this potentially very important interaction in autoimmune disease is still an open question.

It was hypothesized that VitD status and gender may interact to influence the likelihood of developing autoimmunity [53]. VitD metabolites influence peripheral estrogen metabolism. Several authors claimed that there were variations in VitD levels between the sexes, but findings are contradictory because both higher and lower VitD levels were found in women compared with men [54]. In females, low estrogen levels were associated with lower 25(OH)D levels. In contrast, the use of estrogen-containing contraception was associated with a 20% increase in VitD levels. These data are consistent with those obtained in a murine model of MS, showing a gender-specific and estrogen-dependent protective effect of VitD against autoimmunity [54]. A higher prevalence of MS in females and with conditions of VitD insufficiency might support the hypothesis of a protective alliance between VitD and estrogens in humans [55].

Based on preclinical evidence, it is expected that VitD acts as a selective immune regulator. Several genetic and observational studies in humans supported the hypothesis that VitD signaling is involved in the prevention of autoimmune disorders [7]. Unfortunately, cross-sectional and observational studies could not define a causal link between low VitD levels and the risk of developing autoimmunity, as these studies may have been susceptible to confounding factors (comorbidities, anthropometric features, sun exposure, season of collection) and reverse causation effects (low VitD levels as a consequence of reduced sun exposure due to health status instead of predisposing factors to disease). Currently, there is insufficient data from randomized clinical trials to support the protective role of VitD supplementation in decreasing the risk of non-communicable diseases. This is due, at least partially, to debate over the required doses, which may be supraphysiological, and to enrollment biases [53,54].

## 3. Autoimmune Thyroid Disorders

Autoimmune thyroid disorders (AITDs), which include Hashimoto’s thyroiditis (HT) and Graves’ disease (GD), represent the most common autoimmune diseases, with a prevalence of 5% in the general population [33,56]. Both HT and GD are characterized by lymphocytic infiltration of the thyroid gland, which differently affects thyroid function. In HT, the ensuing inflammation causes thyroid follicular cell damage and, eventually, hypothyroidism. Conversely, GD is characterized by thyrocyte hyperplasia (goiter) and excess thyroid hormone synthesis (hyperthyroidism) due to the action of antibodies stimulating the TSH receptor (R) named TSHR-Ab [57].

AITDs are caused by the loss of peripheral (defective self-reactive T cell apoptosis and reduction of Treg function) and central (disrupted thymus deletion of autoreactive T cells) immunological self-tolerance [33]. Like other autoimmune disorders, AITDs are caused by the complex interaction of genetic predisposition and endogenous and environmental precipitating factors.

### 3.1. Hashimoto’s Thyroiditis

In the initial stage of HT, thyroglobulin (Tg) is presented to antigen-presenting cells (APC) and then processed into peptides that bind to T cell receptors. The interaction between CD80/CD86 molecules (on APC) and CD28 (on T cells) represents the second signal required for T cell activation. Studies on T cell phenotypes in HT revealed that the frequency of circulating HLA-DR^+^ (activated) T cells is higher, while CD8^+^ T cells are reduced compared with healthy subjects, but only in the active phase [58]. CD4^+^ T cells predominate in the thyroid infiltrate and many of these are activated [59]. Although in the early stages of HT, a pauciclonal T cell response is plausible, by the time of clinical manifestation, a polyclonal T cell response directed against several autoantigens and epitopes develops. The increased frequency of intrathyroidal Th17 lymphocytes implies a pathogenic role for these pro-inflammatory cells [59]. The role of genetic factors in HT pathogenesis is suggested by the frequent presence of thyroid autoantibodies or other autoimmune disorders in family members of HT probands. Twin studies showed a 0.55 concordance rate in monozygotic but not in dizygotic twins and similar findings were reported for the aggregation of thyroid autoantibodies in the euthyroid twins of individuals with AITDs [60,61,62].

Several genes predisposing to HT emerged, providing insights into the pathogenesis of thyroid diseases, as well as disease co-clustering within families and individuals. It is now known that HT is associated with HLA-DR3 and, to a lesser extent, with HLA-DR4 [63]. Polymorphism of cytotoxic T-lymphocyte-associated protein 4 (CTLA4) was related to an increased risk of 1.3–1.5 of developing HT. GWAS revealed several susceptibility loci shared with other autoimmune disorders, such as the LIM domain containing preferred translocation partner in lipoma (LPP), BTB domain and CNC homolog 2 (BACH2), and MAGI3, which is associated with the progression to hypothyroidism [64,65]. Finally, HT severity was associated with TGF-β polymorphisms. Genetic and epigenetic variants of thyroid-specific genes (Tg and thyroid peroxidase (TPO)) are likely responsible for increased self-peptide binding to HLA and higher stimulation of cellular and humoral responses [66]. Recent studies highlighted the impact of different types of non-coding RNAs (ncRNAs), including microRNAs (miRNAs), long non-coding RNAs (lncRNAs), and circular RNAs (circRNAs) [67,68,69]. Endogenous elements (such as family history, female gender, pregnancy, microchimerism, and gastrointestinal microbiome) and environmental factors (i.g., infection, smoking, stress, micronutrients status, and pollution) might finally trigger the development of autoimmunity, leading to the overt manifestation of HT [70,71].

### 3.2. Vitamin D and Hashimoto’s Thyroiditis

The link between VitD, inflammation, and HT has aroused interest among researchers worldwide [72,73]. Based on the assumption that calcitriol could successfully cooperate with cyclosporine in vitro, Fournier et al. tested this combination in experimental murine models of HT, showing that calcitriol plays a synergistic role in limiting thyroid damage [72]. Animal models that used mice previously sensitized with porcine thyroglobulin showed that VitD effectively decreased the severity of thyroid inflammation or prevented thyroiditis induction compared with control mice [74].

Notwithstanding these encouraging results made by basic science, only a few ambiguous data are available on the effects of VitD status on thyroid function in humans.

Existing data from observational studies point to a higher prevalence of VitD deficiency in patients with HT [75] and to an inverse relationship between circulating levels of VitD and thyroid antibody levels [76,77]. However, results are still inconsistent since, in other studies, the relationship between VitD levels and the TPOAb titer was weak [78] or even absent [79]. Yasmeh et al. obtained contradictory results by observing higher levels of VitD in a cohort of HT women compared with female controls [80], but not in men.

Most of the available randomized control trials demonstrated the ability of VitD to decrease the TPOAb titer [74,81,82,83,84]. Krysiak reported that cholecalciferol effectively reduced TPOAb titers in HT women whose hypothyroidism was well compensated by levothyroxine treatment [82]. A significant decrease in TPOAb titer (−20.3%) after cholecalciferol supplementation (1200–4000 IU/day for 4 months) was observed in 218 euthyroid HT with concomitant VitD deficiency [74]. However, the effect of VitD supplementation on hypothyroidism recovery has not been shown yet [85].

### 3.3. Graves’ Disease

GD shares many features with HT and what determines the type of disorder is a critical issue. GD is the most frequent cause of hyperthyroidism in iodine-sufficient areas [51]. In susceptible individuals, environmental factors could precipitate an organ-specific defect of suppressor T cell functions (HLA-related). Therefore, the suppression of thyroid-specific T helper (Th) cells is defective. In the presence of activated dendritic cells and macrophages, Th cells produce IL-1 and IFNγ, which stimulate the differentiation of B cells to plasma cells and enhance the expression of HLA-DR antigens on the surface of thyroid cells. Dendritic cells and B cells are crucial for the initiation of GD since they express the costimulatory molecules that mediate the interaction between T cells and APC-presenting thyroid antigens. Finally, TSHR-Ab stimulate the TSH-R on the thyroid follicular cells, resulting in increased thyroid hormone production, which may further reduce the number and function of suppressor T cells and stimulate Th cells, thus perpetuating the cyclicity of the disease.

Many studies found a reduction in circulating CD8^+^ T cells and an increase in HLA-DR^+^ T cells in active GD. Tregs largely infiltrate the thyroid gland, but they failed to stop the disorder [63]. From a mutual perspective, thyroid health may coordinate the immunological response, and vice versa, immune system dysfunction may promote the development of thyroid disorders. Solerte et al. reported that NK cell cytotoxicity (NKCC), as well as spontaneous and IL-2-induced TNF-α release, was decreased in NK cells from GD patients compared with controls [86]. Accordingly, cytolytic granule release was reduced in thyrotoxic mice treated with levothyroxine compared with euthyroid mice [87]. Similar results were obtained from the same group in humans [88]. Considering that NK cell activity is affected by age [89], a study compared NKCC in AITD patients with age- and gender-matched healthy controls, demonstrating an impaired NK cell activity in AITDs [90]. As previously outlined, the integration of activating and inhibitory signals regulates cytokine secretion and NKCC. Zhang et al. noticed a decrease in NK cells expressing both activating (NKG2D, NKG2C, NKp30) and inhibitory (NKG2A) receptors in newly onset GD patients compared with matched healthy controls [91]. Additionally, NKG2A^+^ NKs were inversely related to TSR-Ab levels, while NKG2D^+^ NK cells were inversely related to serum-free T4 levels [91], supporting the role of dysfunctional NK cells. Other studies [90,91,92,93,94,95,96], with some exceptions [97,98], agreed that NK cell activity is impaired in GD and observed that antithyroid medication therapy, specifically propylthiouracil, can enhance NK functionality by restoring euthyroidism [99,100].

The pathogenic function of monocytes in GD is poorly understood. Chen and colleagues investigated monocytes in the periphery and thyroid tissue in GD, concluding that CD14^+^CD16^+^ monocytes were significantly expanded in patients with untreated GD and might participate in the production of TSHR-Ab by secreting relatively higher levels of serum-B-cell-activating factor (BAFF). The presence of monocytes was identified in the thyroid tissues of patients with GD, which might maintain and amplify inflammatory responses in situ, and thus, could serve as an important participant in GD pathogenesis [101].

The etiopathogenesis of GD is multifactorial with a genetic component. Studies on twins revealed that monozygotic twins are around 20–30% concordant for GD, which is at least 10-fold higher than dizygotic twins [52]. HLA-DR3 and DQA-1501 are associated with a 2–4-fold increased risk but contribute to only 5–10% of the genetic susceptibility. Polymorphisms of CTLA-4 and other genes that affect lymphocyte response (PTPN22, CD25, FCRL3, and CD226), which are also associated with other autoimmune diseases, confer a higher risk of developing the disease. Polymorphisms in the TSH-R gene are specific for GD and crucial in familial clustering. GWAS revealed many susceptibility loci, including the RNASET2 (rs9355610 tag SNP), FGFR1OP-CCR6 region at 6q27, membrane metalloendopeptidase like (MMEL1), FGFR1OP, prickle planar cell polarity protein 1 (PRICKLE1), and Tg. The β2 adrenergic immunoregulatory factors and the secretoglobulin family 3a member 2 also emerged as susceptible genes [63].

### 3.4. Vitamin D and Graves’ Disease

Data on the effects of VitD status in GD are scant, but the results of the available preclinical studies supported its protective effect. Elocalcitol (BXL-628) reduced IFN-gamma and TNF-alpha-induced CXCL10 protein secretion in human thyrocytes from GD patients more effectively than a placebo (standard anti-thyroid drug treatment). In CD4^+^ T cells, elocalcitol lowered Th1- and Th17-type cytokines and promoted Th2-type cytokine secretion [102]. In addition, VitD insufficiency was associated with the maintenance of hyperthyroidism in GD animal models using mice inoculated with an adenovirus expressing the α-subunit of the thyrotropin receptor, indicating that VitD can control thyroid function [103]. As for HT, a higher prevalence of VitD deficiency was noticed in patients with GD [73].

However, the role of VitD in GD is still an unresolved issue, as the association does not necessarily imply a causal relationship. Though carried out on a limited number of GD and HT participants (255 patients), Inoue et al. suggested that several polymorphisms of the VDR gene may be involved in the development of HT and GD severity, thus supporting epidemiological findings [104]. Disappointingly, the associations between several polymorphisms of the VDR genes and AITD were not replicated in all the studies, especially when performed in different populations (e.g., Asian, Caucasian, and Egyptian).

Kawakami-Tami et al. randomly assigned 30 naïve GD patients to standard methimazole treatment alone or associated with calcitriol observing a greater decrease in serum thyroid hormone levels with the combined therapy [105]. A recent study on a very restricted sample of GD patients failed in demonstrating the implication of DCs in the response of thyroid function to VitD supplementation [106].

The association of selenium and cholecalciferol to anti-thyroid drug treatment was found to be more effective at improving hyperthyroidism compared with methimazole alone in a group of newly diagnosed GD patients with moderate-to-severe hyperthyroidism [107]. Recently, it was suggested that patients with AITD should be tested for VitD and, in the case of deficiency, this should be corrected, ideally by diet, or with VitD compounds at low and safe doses [108].

## 4. Vitiligo

Vitiligo, which is the most common acquired depigmented skin disorder, affects approximately 0.5–2% of the worldwide population; it occurs mainly at a young age without gender predilection. It is characterized by the selective loss of melanocytes, and thus, the patient’s skin presents with milk-white macules and patches [109,110]. According to its clinical features, in 2011, the pathology was classified by the VGICC as SV (segmental vitiligo), NSV (non-segmental vitiligo), and MV (mixed vitiligo). In SV, the depigmentation has an earlier spread that lasts 6-24 months and presents a segmental distribution; furthermore, patients show a positive outcome after melanocyte autografts. NSV is a generic term that includes various pathology forms that present wide macules with a symmetric distribution. MV is characterized by an initial SV that later evolves into NSV [111]. Nowadays, scientists agree with the autoimmune nature of vitiligo, even if the debate about the mechanisms involved in melanocyte death is still open [110]. Vitiligo is often associated with other rare systemic disorders.

Although it is known that the vitiligo outcome and progression are due to melanocyte loss, the mechanisms of melanocyte death are not completely clear. Several options were proposed, such as neural, microvascular, degenerative, melanocyte adhesion, genetic, or autoimmunity theories, but none of these can be considered complete and exhaustive [110,112]. Following the “convergence theory” or “integrated theory”, we might hypothesize that different mechanisms could interact with each other in vitiligo to lead to the destruction of melanocytes, all of which produce similar clinical outcomes [110]. In this hypothesis, specific SNPs in various risk genes create a genetic background that contributes to the development of oxidative stress and immune deregulation, which are two of the crucial factors in the vitiligo outcome. This review focused on the role of innate immunity in vitiligo pathogenesis [113].

### 4.1. The Role of Innate Immunity

In vitiligo, innate immunity can be considered a bridge between oxidative stress and adaptative immunity, which is responsible for melanocyte destruction. It seems that human melanocytes, whose deaths are triggered by oxidative stress, release several damage-associated molecular patterns (DAMPs), such as DNA, HMGB1, and HSP70, that activate innate immunity and, consequently, adaptive immunity. Then, DCs are stimulated and they perform uptake, processing, and MHC presentation to CD8^+^ T cells, with breaking of the tolerance in which IFN-γ has a fundamental role in the initiation of the disease that leads to the destruction of melanocytes [114]. Keratinocytes from lesional skin show an altered expression of several factors, such as stem cell factor (SCF), basic fibroblast growth factor (FGF-2), endothelin-1 (ET-1), and key cytokines (such as IL-1, IL-6, and TNF-α). In particular, several studies reported low levels of SCF and increased expression of TNF-α and IL-6, all of which inhibit melanogenesis [115].

Furthermore, IFN-γ can instruct CXCL9 and CXCL10 chemokine production, which reinforces the recruitment of CD8^+^ T cells, Th1 cells, NK cells, and other ILCs. In the process of chemokine release, a key role is also played by keratinocytes [116].

Interestingly, a transcriptome analysis performed on paired lesional and non-lesional skin biopsies from vitiligo patients and normal skin biopsies from healthy volunteers showed that 17 genes are downregulated (which are mostly related to the melanogenesis process), such as TYRP1, TYR, MLANA, and TRPM1, and 13 genes upregulated, most of which are implicated in innate and acquired immune cells, such as NK and CD8^+^ T cells, and are KLRC1, KLRC2, NKG2D (KLRK1), KLRG1, and others [117].

Several reports showed an increase in the number of circulating NK cells in the blood of vitiligo patients [118,119,120]. However, these altered gene expressions were present in both vitiligo lesional and non-lesional skin, suggesting an altered generalized skin microenvironment that leads to melanocyte death.

Recently, great interest was devoted to the NKGD2 molecule, which is involved in stressed cell recognition by innate immune cells, including NK cells, and the enhancement of TCR signaling. The human ligands for NKG2D consist of distinct stress molecules, such as the major histocompatibility complex (MHC)-class-I-related chain (MIC) A/B and ULBP binding proteins 1–6. Furthermore, NKG2D may have a role in human CD8^+^ T cytotoxicity, as shown by several studies in animal models in which NKG2D signaling promoted the development of long-lasting CD8^+^ T cells with enhanced cytolytic function [121,122,123].

Indeed, melanocyte stress induces MICA and MICB molecules, i.e., NKG2DLs, that interact with NKG2D on CD8^+^ T cells, and this process is thought to be crucial in the development of vitiligo [124] because of results from a mouse model in which NKG2DL expression on target cells is sufficient to induce vitiligo [122]. In this context a relevant pathologic role in vitiligo was conferred to a subset of tissue-resident memory CD8^+^ T cells in lesion areas in the skin of vitiligo patients; these cells moreover present enhanced NKG2D levels and produce increased levels of IFN-γ and TNF-α cytokines [125].

Vitiligo was associated with the APS, which represent a group of clinical conditions characterized by the failure of immune tolerance, functional deterioration of multiple endocrine glands, and the presence of autoantibodies [126]. APS3C was defined as the association between one autoimmune thyroid disease (AITD) and one or more non-thyroidal autoimmune diseases (NTAIDs). In particular, APS3C includes skin autoimmune diseases (including vitiligo), and nervous system autoimmune diseases (such as myasthenia gravis and multiple sclerosis) [127].

Indeed, vitiligo (especially NSV) was found in 20% of patients with other autoimmune diseases. Among these, AITDs are the most frequent: up to 34% of vitiligo patients are affected by thyroid disorders [128]. Moreover, vitiligo is frequently associated with Hashimoto’s thyroiditis and Graves’ disease (for instance, 16.7% of GD patients are affected by other autoimmune diseases, of which 2.6% is vitiligo) [129].

These findings are supported by many studies. A systematic review and meta-analysis by Fan et al. showed that a higher prevalence of anti-thyroperoxidase antibody and anti-thyroglobulin antibody could be observed in vitiligo patients relative to controls, in particular in NSV patients, compared with those with SV [130]. Another interesting study analyzed THAbs titers directed toward thyroxine (T3) and triiodothyronine (T4) in a group of 70 vitiligo patients. It was demonstrated that 95.7% of them had at least one type of THAb, and, in particular, in 50 of them, antibodies against T3 and T4 were present [131]. Another analysis of 123 adults showed that, in most of them, vitiligo appeared after thyroid disease. Moreover, patients with vitiligo had a 2.5-fold higher risk of developing an AITD compared with patients without vitiligo, and this risk increased with age [132]. The reason why THAbs are detected in NTAIDs is that these diseases are characterized by an initial autoimmune inflammation. Another explanation is that inflamed peripheral tissues of NTAID patients (skin in the case of vitiligo) produce molecules that bind thyroid hormones, resulting in new iodinated autoantigens [133], which are then recognized by autoantibodies. Based on these findings, recommending anti-thyroid antibody screening must be considered in all patients with vitiligo in order to diagnose thyroid diseases and improve the patient’s condition.

The correlation between vitiligo and AITDs can be explained by the presence of common heritable susceptibility genes: genome-wide association and linkage analysis techniques identified nine loci [134]. For instance, an autoimmunity susceptibility locus (AIS1) was identified on chromosome 1 in family members affected by vitiligo and Hashimoto’s thyroiditis. At the AIS1 locus, the forkhead transcription factor D3 (FOXD3) was also detected, which seems to be responsible in the case of the concomitant occurrence of vitiligo and an AITD [134].

Other examples are a single nucleotide polymorphism (SNP) on chromosome 8 encoding for Tg, which is shared among patients with vitiligo and an AITD, and the region of IL2RA gene on chromosome 10, whose variants are associated with Graves’ disease (aside from diabetes mellitus type 1, multiple sclerosis, rheumatoid arthritis, and systemic lupus erythematosus) [135].

### 4.2. Vitamin D and Vitiligo

VitD plays an important role in regulating both the innate and adaptive immune systems, but it can also modify the function of other molecules and cells in specific organs, such as the skin. Indeed, VitD [136] influences the tyrosinase enzyme, which is the major enzyme involved in melanin synthesis [137], and the process of melanogenesis in melanocytes via VDR. At the end of the 19th century, it was already demonstrated that VitD might stimulate the differentiation of immature melanocyte precursors. Oikawa and Nakayasu [138] showed that melanotic mouse melanoma cell lines treated with VitD3 displayed an improvement in tyrosinase activity and melanogenesis. Moreover, Tomita et al. showed that a 6-day VitD3 treatment increased the tyrosinase content of cultured human melanocytes [139]. Moreover, it was reported that the active form of VitD can reduce apoptosis in keratinocytes and melanocytes via the production of IL-6 and sphingosine-1-phosphate [140]. In 2018, Tang et al. studied a new mechanism of VitD that protects melanocytes: the activation of the WNT/β-catenin pathway [141]. Indeed, VitD needs the β-catenin pathway in order to modulate downstream targets such as MITF (melanocyte-inducing transcription factor) and apoptotic pathways. This study also demonstrated that vitiligo patients were characterized by higher MDA (malondialdehyde), which is a product of lipid peroxidation [142] and lower VitD levels compared with healthy controls) [141]. Therefore, these findings showed the potential antioxidant effect of VitD.

Low serum VitD levels characterize many autoimmune disorders, such as AITDs, MS, and vitiligo. The first systematic review about the relationship between VitD levels and vitiligo by Upala et al. showed that vitiligo is significantly associated with a low serum VitD concentration [143]. A case–control study on 40 vitiligo patients and 40 healthy controls demonstrated that one vitiligo patient (2.5%) and the majority of healthy controls (82.5%) had sufficient serum 25(OH)D levels, whereas 97.5% of patients and 12.5% of controls had deficient 25(OH)D levels. Based on these findings, VitD level screening may be recommended to all patients with vitiligo for possible VitD supplementation [144]. However, until now, some studies showed statistically insignificant differences in VitD levels between vitiligo patients and healthy controls: VitD levels are usually low in both groups [145,146].

El-Hanbuli et al. demonstrated that vitiligo melanocytes were characterized by a significant reduction in VDR protein in comparison to normal melanocytes. It was observed that in patients affected by vitiligo, VDR expression was remarkably reduced in the areas of “lesional/perilesional” skin, while re-pigmentation was accompanied by an increase in VDR expression [147].

Vitiligo pathogenesis and etiology are very complex because of the high number of processes and cells that are involved. Moreover, many studies about the association between VDR gene polymorphisms and susceptibility to vitiligo provide interesting results. An analysis by Aydıngöz et al. in 2012 reported that individuals with TaqI polymorphism had a 2.23-fold increased risk of developing vitiligo [148]. In 2015, a meta-analysis by Li et al. showed that ApaI or BsmI gene polymorphism may increase the risk of vitiligo in East Asian populations [149]. However, a more recent analysis showed divergent results. In 2020, Katsarou et al. demonstrated that there was no evidence to support an association between susceptibility to vitiligo and VDRBsmI, TaqI, and FokI polymorphisms [150]. Furthermore, Saudi et al. in 2021 showed that VDR polymorphisms were not correlated with vitiligo, even though a particular genotype allele appeared to be more expressed in vitiligo patients [151]. Since VitD positively influences the immune system and it stimulates the process of melanogenesis in vitro, the question whether increased VitD levels could improve vitiligo condition was raised. Narrow-band ultraviolet B radiation (NB-UVB) was found to be the gold standard in the treatment of vitiligo. In 2018, a study by El-Hanbuli et al. showed that NB-UVB phototherapy is associated with increased VDR expression and VitD synthesis. Moreover, these effects are correlated with a better re-pigmentation response [147]. Ibrahim et al. suggested that in vitiligo patients, cumulative doses of NB-UVB could increase VitD levels after 12 and 24 weeks of therapy [152]. According to the previous study, NB-UVB radiation can play a significant role in inducing re-pigmentation [152].

However, the specific molecular mechanism is still not clearly understood, and thus, further studies are needed. An observation conducted on 101 vitiligo patients showed that after six months of follow-up, sufficient 25(OH)D levels were associated with the stability of the disease and re-pigmentation. This evidence suggested a possible beneficial role of VitD supplementation on vitiligo patients [153]. However, until now, there are still few data about this treatment approach.

## 5. Multiple Sclerosis

Autoimmune demyelinating diseases can affect the nervous system at the level of peripheral nerves, ranging from acute inflammatory demyelinating polyneuropathy to chronic forms, and at the level of the central nervous system (CNS), with MS being the most prominent one [154].

MS is an autoimmune disease that affects the CNS and is characterized by chronic inflammation, perivascular infiltrates of mononuclear inflammatory cells, gliosis, demyelination of nerves, and the formation of multiple plaques in the spinal cord and brain, ultimately resulting in nerve damage that leads to various levels of neurological signs and symptoms and disabilities. It is a debilitating condition that presents early in life, with a mean age of onset of 28–31 years of age [155]. The prevalence of the disease varies in different regions of the globe, ranging from 15/100,000 to 250/100,000 [156]. As of today, the primary cause of MS remains unclear. MS is a complex disease and the causes of its occurrence can be endogenous and exogenous; it was shown that several genes can increase susceptibility to MS, in addition to several environmental factors, in particular a lack of vitamin D or ultraviolet B light (UVB) exposure, infection with some viruses (with Epstein–Barr virus (EBV) being the most likely candidate for a causative virus), obesity, and smoking [157]. In this regard, a recent report strongly suggested EBV infection as the leading cause of MS [158].

Epidemiological studies indicated that MS is prevalent in high-latitude countries and this phenomenon is associated with a higher risk of vitamin D deficiency [159]. The occurrence of MS is more common in females, with the female-to-male ratio nowadays reaching close to 3:1 in most developed countries [160].

In the earlier phases of the disease, perivenular inflammatory lesions are markedly present and consist of mononuclear infiltrations. Those inflammatory lesions further lead to the formation of demyelinating plaques, which is the distinguishing pathological hallmark of MS [161].

The inflammatory infiltrates contain T lymphocytes, which are dominated by MHC class I restricted CD8^+^ T cells. The presence of B cells and plasma cells was also recorded, although in much lower numbers. Oligodendrocyte damage and demyelination occur as a result of inflammation. Axons are relatively preserved in the early stages of the disease; however, as the disease progresses, irreversible axonal damage develops [162,163].

The traditional view describes MS as the pathology mediated by autoreactive T cells recognizing auto-antigens that are activated in the periphery by mechanisms yet to be unveiled, such as molecular mimicry and the breakdown of immunological tolerance. The characteristic neuronal damage in MS results from a complex and dynamic interplay between the immune system (adaptive and innate), glia (myelin-making oligodendrocytes and their precursors, microglia, and astrocytes), and neurons. It has been confirmed for a few decades now that the CNS is not an immunologically privileged site [164] since the afferent and efferent pathways of communication between the CNS and the immune system were documented [165]. Antigens can exit from the brain and can induce antibody production in the cervical lymph nodes in the presence of activated lymphocytes. Activated T cells proliferate and express several adhesion molecules (the alpha 4 integrin being one of the key players) and receptors and secrete metalloproteases and pro-inflammatory mediators, allowing them to activate the brain–blood barrier (BBB) and interact with it and enter the brain. Once in the brain, T cells undergo reactivation by the local autoantigens presented by MHC class II molecules expressed on activated astrocytes, macrophages, and microglia [165,166].

During the progression of the disease, in addition to T cells (both CD8^+^ and CD4^+^), NK cells, dendritic cells, and B cells also infiltrate the CSF and then into CNS tissue, orchestrating local inflammation [167].

Conventionally, MS was considered an autoimmune disease in which the adaptive immune system has a critical role. This is mainly due to the numerous studies done on animals via the usage of experimental autoimmune encephalomyelitis (EAE). In addition, early analysis of humans was done only by analyzing the cerebrospinal fluid and peripheral blood, and thus, the conclusion about the predominant role of the adaptive immune system, in particular T cells, was expected. Both helper (CD4^+^) and cytotoxic (CD8^+^) T cells were described in MS lesions: CD4^+^ T cells are more predominantly present in the perivascular space, while CD8^+^ T cells are distributed within the parenchyma [168].

Nevertheless, the important roles of B cells as a neglected component of the adaptive immune system in this pathology and the innate immune system were revealed. Owing to the early and rapid success of B-cell-depleting antibodies in limiting MS lesion formation, there is renewed attention on the role of B cells. Namely, there is documented evidence of a positive regulatory role for a subset of B cells in MS disease that can release several inhibitory cytokines, such as IL-10, TGF-β, and IL-35 [169]. Furthermore, it was shown that functional B regulatory cells (Breg) can impair proinflammatory cytokine production by DCs, favour Treg cell expansion, and inhibit the development of T effector cells [170].

Participation of the innate immune system in MS pathogenesis has been under extensive investigation. Macrophages (classically and alternatively activated) originating from peripheral blood can infiltrate active MS lesions and remove myelin debris and inflammatory derivatives. Microglial cells are the primary endogenous phagocytes of the CNS, and their abundant presence was confirmed in MS lesions. Nevertheless, the exact nature of their predominant role (pathogenic and/or protective) remains ill-defined [171].

At the same time, NK cells are also much studied for their possible role in controlling MS pathogenesis, although the effective role of NK cells in MS remains a subject without conclusive results. In this context, Morandi et al. demonstrated that in healthy subjects, CD56^bright^ NK cells work ex vivo as inhibitory cells on CD4^+^ T cells via adenosine release, whereas in the context of autoimmune diseases, such as in juvenile idiopathic arthritis patients, these innate immune cells failed to perform this effect and, at the same time, CD56^bright^CD16^−^ NK cells derived from inflammatory pleural effusions had a marked inhibitory function, emphasizing the possible regulatory role played by this subtype of NK cells [172]. Adenosine is an important immunosuppressive molecule with potentially relevant roles in autoimmune diseases [173].

More recently, Laroni et al. added new data on the functioning of the NK subset by identifying a CD56^bright^CD16^−^ NK cells subset that was able, upon inflammatory cues, to kill autologous CD4^+^ T cell proliferation via granzyme B release and NKp30 and NKp46 involvement [174], thus reinforcing the notion that this subset of NK cells could be associated with the regulation and control of MS. Of note, the treatment of anti-CD25 mAb (daclizumab) in MS patients resulted in the expansion of circulating CD56^bright^ NK cells [175].

Other components of the immune innate system, such as astrocytes or complements, are activated by cytokines and chemokines that are secreted in the early phases of an inflammatory reaction. This activation leads to a direct attack on myelin, axons, and glia, and it is mediated by cytotoxic cells and phagocytosis, proteases, cytokines, complement, glutamate, NO, and other intermediates.

### Vitamin D and Multiple Sclerosis

VitD effects on immune response in MS patients were confirmed in many in vitro studies, and numerous studies in animal models of MS and EAE strongly suggested the protective role of VitD. Despite this, the results of a few clinical trials that explored the influence of VitD on immune response in patients with MS were contradictory.

The abundant presence of CD8^+^ T cells in acute and chronic plaques is well documented, as well as the oligoclonal expansion of these cells in CSF and the peripheral blood of MS patients [176,177].

CD8^+^ T cells have a higher expression of VDR than CD4^+^ T cells [178]. Lysandropulos et al. found that 1,25(OH)2D3-treated CD8^+^ T cells secreted lower quantities of the pro-inflammatory cytokines IFN-γ and TNF-α and more anti-inflammatory cytokines, such as IL-5 and TGF-β [179].

Concerning animal models of MS and EAE, VitD exerts formidable effects on Th1 and Th2 cells and Th1/Th2 cytokine synthesis, thus inhibiting EAE induction [180,181].

Pedersen et al. showed that 1,25(OH)2D3 treatment significantly reduces the clinical severity of EAE by decreasing monocyte trafficking and chemokine levels [182], while Mayne et al. showed that VitD acts directly on pathogenic CD4^+^ T cells through the VDR expressed in the nucleus of these cells, leading to the inhibition of EAE [183].

Conversely, numerous studies carried out on MS patients reported conflicting results.

Functional deficiencies in Treg cells with markedly decreased production of IL-10 were described in the peripheral blood of MS patients in several studies [184,185,186]. Smolders et al. demonstrated improved CD25^+^CD4^+^ Treg cell function associated with high levels of 25-OHD in serum in MS patients, while this association was not confirmed after vitamin D3 supplementation [187].

In an EAE mouse model, it was shown that VitD supplementation can lead to an increase in the differentiation rate of Treg cells while inhibiting the proliferation of Th1 and Th17 cells [188].

The effect of VitD on B cells is still not completely clear. In vitro studies showed that 1,25(OH)2D3 has several effects on B cells that may be beneficial in MS, such as inhibition of B cell proliferation and plasma cell differentiation, induction of apoptosis, augmentation of the generation of B memory cells, enhancement of Breg cell activity, and participation in immunoglobulin production. Moreover, it was shown that VitD promotes IL-10 production in human B cells; however, the inhibitory effects of vitamin D were not confirmed in vivo [189].

Dysregulation of the phenotype and/or function of NK cells was reported in various autoimmune disorders through multiple mechanisms [190], and we previously mentioned the potential role of NK cells in regulating GD [191].

Moreover, in MS, NK cells could have a role in the regulation of the autoimmune process, mainly via cytokine secretion or direct cytotoxic activity on effector cells, such as autoreactive T cells or antigen-presenting cells (APCs) [192,193]. Actually, in MS, there are reports that show that NK cells also have the ability to kill oligodendrocytes, astrocytes, and microglia via the NKG2D receptor, also suggesting a negative effect of these cells on tissue injury in MS [194].

Concerning VitD and NK cells, it was reported that the treatment of EAE-induced SJL mice with VitD3 or with monomethyl fumarate (MMF) contrast can prevent EAE, and this process was associated with enhanced in vitro NK cell lysis of monocyte-derived dendritic cells [195], corroborating previous data from the same group [196].

Of note, other studies reported that VitD3 ameliorates the clinical score in EAE mice [197], and others showed that this treatment was able to downregulate several proinflammatory cytokines, such as IL-6 and IL-17 [198].

Taken together, these data support the fact that VitD actively modifies the immune response either by acting directly on T, B, and NK cells or by modulating DC function. Henceforward, it is appropriate to presume that a deficiency in VitD could augment the risk and/or progression of MS. However, that might not be the case.

Several studies confirmed that 25(OH)D levels are lower in MS patients (20–84%) than in healthy controls [55,199,200,201].

Interestingly, lower VitD levels were found in MS patients with progressive forms of the disease with increased disability [199,200]. Munger et al. showed that higher serum levels of 25(OH)D were associated with a significantly lower risk of incident MS, but only in Caucasians and not in Afro-Americans or Hispanics, and particularly in people before 20 years of age [202]. Furthermore, the same group demonstrated that women who ingested more than 400 IUs of vitamin D per day had a 41% decreased risk of developing MS [203].

Conversely, two studies demonstrated that the administration of cholecalciferol and/or alfacalcidol to MS patients had no effect or did not lead to significant improvement in clinical symptoms of the disease and the number of relapse episodes [204,205]. A double-blind, placebo-controlled, randomized trial showed no difference between RRMS patients that were receiving either a low dose (1000 IU/daily) or high dose (6000 IU/daily) over six months in terms of clinical and MRI parameters [206]. Likewise, several other studies did not find a positive correlation between VitD supplementation and improvement in clinical parameters in MS patients [207,208,209].

A published meta-analysis regarding the beneficial effect of VitD as a treatment for MS gave inconclusive suggestions, although it is worth noting that a higher level of VitD tends toward providing a favorable clinical response [210,211,212]. In conclusion, numerous studies suggested that VitD supplementation may be protective against MS. Further research is needed to establish how and when individuals with MS should be supplemented with VitD.

## 6. Conclusions and Future Perspectives

Genetic and biochemical data soundly demonstrated that VitD regulates a very large number of genes, most of which are not implicated in bone homeostasis. VitD status has a strong genetic basis, as emerged in twin studies. Large GWAS revealed at least four genes implicated in determining VitD levels and action. VitD’s influence on the immune system is supported by numerous studies in animal models and in vitro. The recent results of the VITAL trial demonstrated a protective role of VitD supplementation on autoimmune disorders. However, there is still no consensus on the extraskeletal effects of VitD on human health and diseases, mainly due to the lack of evidence of the clinical effect of VitD supplementation in the case of deficiency. This is likely due to the poor quality of randomized control trials, which were not homogeneous in VitD treatment (dose, molecule, treatment scheme), the definition of VitD status (VitD levels, VitD assay), and study population (sample size, age, sex, enrollment biases). Still, factors regulating local VitD activity are only partially understood. Standardization of the definition of VitD status, treatment scheme, and VitD assay is warranted to definitively clarify the implication of VitD in autoimmune disorders.

## Figures and Tables

**Figure 1 ijms-24-04689-f001:**
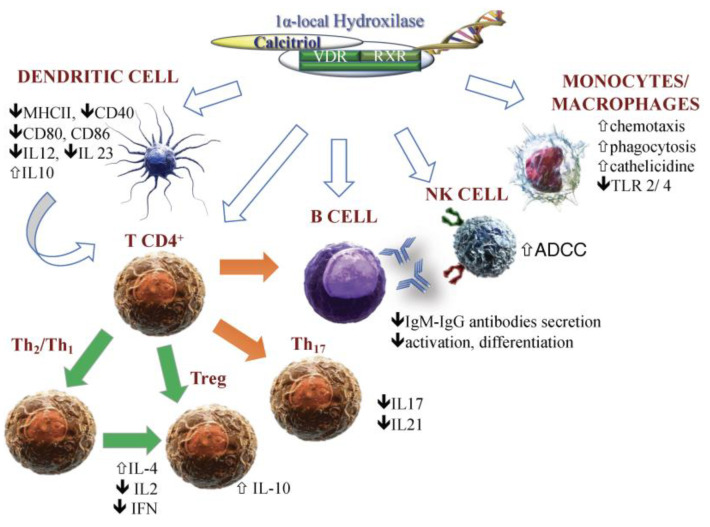
Vitamin D and immune cells crosstalk. Vitamin D (calcitriol) directly and indirectly influences and regulates both innate and adaptive immune cells, which widely express the vitamin D receptor (VDR). RXR—retinoic acid receptor; NK—natural killer cells; ADCC—antibody-dependent cell-mediated cytotoxicity; IL—interleukin; MHCII—major histocompatibility complex class II; Th—T helper; TLR—toll-like receptor; green arrow—stimulation; orange arrow—inhibition.

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
