# Peer review of "How Does Vitamin D Affect Immune Cells Crosstalk in Autoimmune Diseases?"

_ijms, 2023, doi:10.3390/ijms24054689_

Round 1
Reviewer 1 Report
This is a nice and comprehensive review. Different autoimmune diseases are highlighted and the role of Vitamin D within each is discussed. The coverage was thorough.
Author Response
We thank Reviewer 1 for her/his comments. For English, we thoroughly rechecked the whole text and corrected several errors, which are evident in the red or yellow version.
Reviewer 2 Report
Major comments:
1. As clearly explained, autoimmune thyroid disorders consist of two distinct types of disorders: Hashimoto’s thyroiditis (HT) and Graves’ disease (GD). Therefore, the paragraph that focuses on genetic predisposition to AITD should be rewritten to differentiate GWAS studies conducted on HT from those performed on GD, as we know that HT and GD have partly overlapping but mostly different genetic background.
2. In line with the previous comment, it is vital to review differences between these two diseases in the light of immunological background and the role of VitD.
3. The sentence between lines 220-222 (The identification of polymorphisms in genes involved in VitD metabolism in cohorts with 221 AITD increases the likelihood that VitD status is implicated in the development of the disease [85]) relies solely on results from one candidate-gene study that used very limited number of participants. Neither of these genes were found to be involved with HT or GD in GWAS studies (please search the GWAS catalog). Therefore, the significance of these genes with AITD is not supported with genetic data and this sentence shoule be changed accordingly.
4. Current literature data still do not agree if VitD deficiency is the result of HT disease or a part of its cause. A paragraph reflecting on this problem is needed in this manuscript.
Minor comments:
1. The correct reference for the statement in lines 210-213 is not 82, which is a review paper, but this one: Fournier C, Gepner P, Sadouk M, Charreire J (1990) In vivo benefcial efects of cyclosporin A and 1,25-dihydroxyvitamin D3 on the induction of experimental autoimmune thyroiditis. Clin Immunol Immunopathol 54:53–63
Author Response
We thank Reviewer 2 for the different points put forward because they allowed us to significantly improve our article that had some weaknesses and some unclear parts. Therefore below are point by point corrections made to the text in accordance with the reviewer's suggestions and criticisms.
As clearly explained, autoimmune thyroid disorders consist of two distinct types of disorders: Hashimoto’s thyroiditis (HT) and Graves’ disease (GD). Therefore, the paragraph that focuses on genetic predisposition to AITD should be rewritten to differentiate GWAS studies conducted on HT from those performed on GD, as we know that HT and GD have partly overlapping but mostly different genetic background
R) We thank Reviewer 2 for her/his comments. We split section 3 of the original manuscript “ Autoimmune thyroid disorders” into two separate subparagraphs: 3.1 Hashimoto’s thyroiditis and 3.3 Graves’disease. Thus, in this revised version of the manuscript, genetic predisposition was treated separately for Hashimoto’s thyroiditis (page 5, lines 226-246) and Graves’ disease (page 7, lines 331-341) as you suggested.
- In line with the previous comment, it is vital to review differences between these two diseases in the light of immunological background and the role of VitD.
R) We thank Reviewer 2 for her/his comments. The pathogenesis of Graves' disease and Hashimoto's thyroiditis was more fully addressed in this revised version of the manuscript (page 5, lines 214-224 and page 7 lines 288-239). Accordingly, vitamin D status and supplementation's potential roles were described more clearly in two separate subparagraphs: “3.2 Vitamin D and Hashimoto’s thyroiditis” (page 6, lines 252-283 of the revised version of the manuscript) and Vitamin D and “3.4 Vitamin D and Graves’ disease” (page 8, lines 344-377 of the revised version of the manuscript) as you suggested.
- The sentence between lines 220-222 (The identification of polymorphisms in genes involved in VitD metabolism in cohorts with 221 AITD increases the likelihood that VitD status is implicated in the development of the disease [85]) relies solely on results from one candidate-gene study that used very limited number of participants. Neither of these genes were found to be involved with HT or GD in GWAS studies (please search the GWAS catalog). Therefore, the significance of these genes with AITD is not supported with genetic data and this sentence should be changed accordingly.
R) We thank Reviewer 2 for her/his comments. The sentence was corrected and rephrased according to your suggestion “Though carried out on a limited number of GD and HT participants (255 patients), Inoue et al. suggested that several polymorphisms of VDR gene may be involved in the development of HT and GD severity thus supporting epidemiological findings [98]. Disappointingly, the associations between several polymorphisms of the VDR genes and AITD were not replicated in all the studies, especially when performed in different populations (e.g., Asian, Caucasian, Egyptian) (82).”
- Current literature data still do not agree if VitD deficiency is the result of HT disease or a part of its cause. A paragraph reflecting on this problem is needed in this manuscript.
R) Your comment was really appreciated. Since this is an open and crucial issue not only for Hashimoto’s thyroiditis but, more generally, for all autoimmune disorders (and not only) we added a comment in the section 2. Vitamin D and immune regulation” (page 5, line 185-193 of the revised version of the manuscript).
Unfortunately, cross-sectional and observational studies could not define a causality link between low VitD levels and the risk of developing autoimmunity as these studies may be susceptible to confounding factors (comorbidities, anthropometric features, sun exposure, season of collection) and reverse causation effects (low VitD levels as a consequence of reduced sun exposure due to health status instead of predisposing factors to disease). Currently, there is insufficient data from randomized clinical trials, to support the protective role of VitD supplementation in decreasing the risk of non-communicable diseases. This is due, at least partially, to debate over the required doses, which may be supraphysiological, and to enrollment biases [51, 52].
Minor comments:
The correct reference for the statement in lines 210-213 is not 82, which is a review paper, but this one: Fournier C, Gepner P, Sadouk M, Charreire J (1990) In vivo benefcial efects of cyclosporin A and 1,25-dihydroxyvitamin D3 on the induction of experimental autoimmune thyroiditis. Clin Immunol Immunopathol 54:53–63
R) We thank the Reviewer. Reference “Fournier C, Gepner P, Sadouk M, Charreire J (1990) In vivo beneficial effects of cyclosporin A and 1,25-dihydroxyvitamin D3 on the induction of experimental autoimmune thyroiditis. Clin Immunol Immunopathol 54:53–63” was added (ref. #72 of the revised version of the manuscript).
Reviewer 3 Report
It is a topical and interesting review.
I suggest addressing the role of the IL-33/ST2 axis and vitamin D in autoimmune diseases and "synthetically" expanding the aspects regarding vitamin D, gender and autoimmunity (lines 84 and 85).
pay attention to some oversights in the text and to the citation of some references (ex. lines 50, 80,....)
Author Response
We thank Reviewer 3 for her/his suggestion regarding to discuss the role of the IL-33/ST2 axis and vitamin D in autoimmune diseases. We then added a few sentences in the article highlighting the potential role of IL-33 in the autoimmune diseases as well as the role of gender. These sentences can be found highlighted in yellow in the highlighted corrections version. Regarding possible errors in the text and references, we carefully rechecked the whole text and citations and corrected several errors.
Round 2
Reviewer 2 Report
Thank you for answering to all of my comments. I have no further comments.